# The Association between Prenatal Exposure to Per- and Polyfluoroalkyl Substances and Respiratory Tract Infections in Preschool Children: A Wuhan Cohort Study

**DOI:** 10.3390/toxics11110897

**Published:** 2023-11-02

**Authors:** Haiyun Huang, Xiaojun Li, Yican Deng, Siyi San, Dongmei Qiu, Xiaoyu Guo, Lingyun Xu, Yang Li, Hongling Zhang, Yuanyuan Li

**Affiliations:** 1School of Medicine and Health, Wuhan Polytechnic University, Wuhan 430023, China; amyhuang9912@gmail.com (H.H.); cosmosongf@163.com (Y.D.); 13419698744@163.com (S.S.); qdm1232023@163.com (D.Q.); 2Key Laboratory of Environment and Health, Ministry of Education & Ministry of Environmental Protection, and State Key Laboratory of Environmental Health (Incubation), School of Public Health, Tongji Medical College, Huazhong University of Science and Technology, Wuhan 430030, China; lixiaojun@hust.edu.cn; 3School of Life Science and Technology, Wuhan Polytechnic University, Wuhan 430023, China; 17683931781@163.com (X.G.); doctorxly9898@163.com (L.X.); yangli@whpu.edu.cn (Y.L.)

**Keywords:** per- and polyfluoroalkyl substances (PFASs), respiratory tract infections (RTIs), pregnant women, cord blood, preschool children, cohort study

## Abstract

This study investigates the association between prenatal exposure to per- and polyfluoroalkyl substances (PFASs) and the incidence and frequency of respiratory tract infections (RTIs) in preschool children. We selected 527 mother–infant pairs from Wuhan Healthy Baby Cohort (WHBC), China. Ten PFASs were measured in umbilical cord serum, and we collected data on common RTIs in preschool children aged 4 years through a questionnaire. Associations of single PFASs with the incidence and frequency of RTIs were analyzed via Logistic regression and Poisson regression, while the collective effect was assessed by weighted quantile sum (WQS) regression. Furthermore, stratified and interaction analyses were performed to evaluate if there were sex-specific associations. We found a positive correlation between perfluorododecanoic acid (PFDoDA) and the incidence of tonsillitis, with several PFASs also showing positive associations with its frequency. Moreover, perfluorotridecanoic acid (PFTrDA) showed a positive link with the frequency of common cold. The results of WQS regression revealed that after adjusting for other covariates, PFASs mixture showed a positive association with the incidence of tonsillitis, the frequency of common cold, and episodes. In particular, perfluoroundecanoic acid (PFUnDA), PFDoDA, PFTrDA, perfluorodecanoic acid (PFDA) and 8:2 chlorinated polyfluorinated ether sulfonic acid (8:2 Cl-PFESA) had the most significant impact on this combined effect. The results suggest that both single and mixed exposures to PFASs may cause RTIs in preschool children. However, there was no statistically significant interaction between different PFASs and sex.

## 1. Introduction

Per- and polyfluoroalkyl substances (PFASs), a series of fluorinated synthetic organic compounds, have been widely used in the production of household cleaning products, textiles, and fire-fighting formulas for industrial and consumer use [1]. Due to their exceptional stability and persistence in the environment, PFASs are difficult to degrade and have an extended biological half-life in the human body. Notably, perfluorooctane sulfonic acid (PFOS) and perfluorooctanoic acid (PFOA) have been successively recognized as persistent organic pollutants and are subject to global regulation [2]. Currently, chlorinated polyfluorinated ether sulfonic acids (Cl-PFESAs, trade name F-53B) are used as a substitute for PFOS in China. Numerous studies have shown that PFASs can be efficiently transferred across the placenta, resulting in fetal exposure to PFASs in utero through blood exchange with the mother [3]. During this critical period of growth, there is evidence of reduced birthweight [4,5], neurobehavioral development [6,7], weakened immune responses [8,9], liver function [10], and increased adiposity in newborns and children [11,12].

Respiratory tract infections (RTIs) are a syndrome mainly characterized by symptoms such as fever, cough and sore throat. They are very common in primary care settings and are a major cause of illness worldwide. Among these infections, lower RTIs are the primary cause of death in children under 5 years old [13]. Meanwhile, in China, children below 5 years old experience upper RTIs among children [14].

The effects of PFASs exposure on RTIs in children have been inconsistent. Some studies have reported a positive relationship between PFASs exposure and RTIs [9,15,16,17]. Conversely, a Norwegian birth cohort found a negative association between PFOS and PFOA exposure and the risk of common colds in children aged 0–3 years, while perfluorohexane sulfonate (PFHxS) exposure was correlated with a surge in the risk of bronchitis, pneumonia, and throat infection [18]. However, findings from a Danish birth cohort revealed no association between PFOS and PFOA and bronchitis, pneumonia, and other infections at 18 months of age [19].

In reality, human exposure involves a complex mixture of environmental chemicals, rather than a single substance. Therefore, risk assessments that focus on specific compounds may not capture the full health risks of exposure to a spectrum of chemicals. In addition, most studies have focused on conventional PFASs, leaving a noticeable gap in understanding the association between novel PFASs alternatives and respiratory infections in children.

Therefore, based on a large birth cohort, this study analyzes the single and combined effects of 10 types of PFASs on the incidence and frequency of RTIs in children. The findings will provide a valuable reference for further health risk assessment.

## 2. Materials and Methods

### 2.1. Study Population

The subjects of this study were derived from the data of the birth cohort study conducted at the Wuhan Maternal and Child Health Hospital (WMCHH), China. Participants were considered for the study based on the criteria: (1) early pregnancy (<16 weeks); (2) singleton pregnancy; (3) permanent residents of Wuhan with no recent plans to relocate; (4) intending to have antenatal care and childbirth at the recruiting hospital; (5) consent to personal follow-up and to provide biological samples. Finally, 527 mother–child pairs were included. All women enrolled in the study gave informed consent. The research was approved by the ethics committee of WMCHH (#2010009, 9 October 2012) and Huazhong University of Science and Technology ([2012]#14, 8 October 2012).

### 2.2. PFASs Measurement

After extraction by the ion-pair liquid–liquid extraction method, PFASs in the umbilical cord serum were analyzed by the Acquity Ultra Performance Liquid Chromatography coupled to a Xevo TQ-S triple quadrupole mass spectrometer (Waters, Milford, MA, USA) [20]. A total of 10 PFASs were detected, including 6 perflfluorocarboxylic acids (PFCA), 2 perfluorosulfonic acids (PFSAs), and 2 Cl-PFESAs. The chemicals, molecular formulas, and structures can be found in Appendix A. For PFASs exposure levels detected below the limit of quantification, the square root of LOQ divided by 2 was used as a substitute. Levels below the LOQs were replaced by the LOQs divided by the square root of two in further association analysis [20].

### 2.3. RTIs Measurement

This study focused on four prevalent RTIs in preschool children: common cold, bronchitis, pneumonia, and tonsillitis. To collect information on these illnesses during the previous 12 months, standardized questionnaires were distributed to parents in collaboration with kindergarten healthcare providers when the children were 4 years old. Parents were asked to check the four options to indicate whether their child had suffered from these illnesses in the past year, and, if so, to note the number of incidences. However, some parents did not complete this section, resulting in missing data for 12, 8, 4, and 7 individuals, respectively.

### 2.4. Potential Confounders

We selected covariates based on literature reports and variables causing a change of 10% or more in the regression coefficient, including maternal age (continuous variable), family annual income (<CNY 100,000; ≥CNY 100,000), child sex (male; female), gestational week (continuous variable), parity (primipara; multipara), breastfeeding duration (<6 months; ≥6 months), and child exposure to passive smoking (yes; no). These potential confounders were adjusted for in the regression model.

### 2.5. Statistical Analyses

First, we described the distribution of the basic characteristics of the subjects. Since the PFASs concentrations showed a right-skewed distribution, a logarithmic transformation was applied to approximate a normal distribution, to facilitate subsequent analysis. The central and dispersion trends of PFASs concentrations were described by geometric mean (GM) and percentiles (P5, P25, P50, P75, and P95).

Second, we employed a single-pollutant model to analyze the association between individual PFASs exposure and RTIs. PFAS concentrations were included in the model as independent variables, both as continuous variables (with log2 transformation) and as categorical variables (divided into low, medium, and high concentration groups based on tertiles, with the low concentration group serving as the reference). We treated the incidence of disease as a binary variable to analyze the association using a Logistic regression model and a Poisson regression model, incorporating the number of illnesses as count data, was used to explore the relationship between PFASs exposure and the frequency of RTIs in preschool-aged children. In addition, we also assessed the relationship in sex-stratified subgroups and examined potential interactions by including interaction terms in the models.

Third, WQS regression was used to analyze the association between PFASs mixture exposure and RTIs. A weighted WQS index was measured to assess how each PFASs contributed to the combined effect of the mixture. Because WQS regression assumed a one-way relationship between mixture exposure and the outcome, both positive and negative associations were set in separate regressions to accurately analyze the direction of the relationship.

The following sensitivity analyses were conducted: (1) excluding preterm, low birth weight, and macrosomic infants, we reran the models; (2) to explore the stability of the WQS results, Quantile g-computation was used to verify the weights of various PFASs in the combined effect of PFAS mixed exposure.

WQS regression and Quantile g-computation were analyzed using the gWQS and qgcom packages in R software (Version 4.1.3), and other analyses were conducted in SAS software (Version 9.4).

## 3. Results

### 3.1. Sample Characteristics

Of the 569 participants with PFASs data from March 2014 to February 2015, five children were excluded for congenital malformations. Ultimately, 527 participants had data on RTIs and were incorporated in this study (Appendix A).

Appendix A presents the distribution of selected features in the population. The average age (mean ± SD) of participant women was 29.12 ± 3.14 years, within the normal BMI range, and had at least a junior school degree. Their annual family income was less than CNY 100,000. None of the pregnant women actively smoked or drank alcohol, but 27.13% of the women and 47.82% of the children were exposed to secondhand smoke. Most of the children were firstborns, and 53.89% were male.

The gestational age was about 40 weeks, the preterm birth rate was 3.23%, and the rates of low birth weight and macrosomia were 1.71% and 5.31%, respectively. A total of 65.84% of mothers breastfed for 6 months or more. Data on children’s RTIs were collected when they were about 4 years old.

### 3.2. Distribution of PFASs

The quantification limits, detection rates, and concentration distributions of the 10 PFASs in cord serum are shown in Table 1. The detection rates for all PFASs were over 90%, with perfluorododecanoic acid (PFDoDA) having the lowest detection rate of 90.89%. PFOS, PFOA, and 6:2 chlorinated polyfluorinated ether sulfonate (6:2 Cl-PFESA) had the highest concentrations, with median concentrations of 4.17 ng/mL, 1.62 ng/mL, and 0.79 ng/mL, respectively. Concentrations of the other PFASs had median values between 0.03 and 0.42 ng/mL.

### 3.3. Associations between Single PFASs and RTIs

In the single pollutant model, subjects were grouped into three categories based on tertiles of PFAS concentrations: low, medium, and high. Using the low concentration group as a reference, we conducted both Logistic and Poisson regression analyses (Table 2 and Table 3).

For the common cold, the medium groups for PFTrDA (OR = 2.00, 95% CI: 1.18, 3.41) and PFHxS (OR = 1.68, 95% CI:1.00, 2.91) showed a correlation with an increased incidence of incidence (Table 2). A correlation with an increased frequency of the common cold was observed for the high concentration of PFUnDA (β = 0.16, 95% CI: 0.01, 0.31), both medium and high concentrations of PFDoDA (medium concentration β = 0.29, 95% CI:0.14, 0.45; high concentration β = 0.22, 95% CI: 0.06, 0.37), and PFTrDA (medium concentration β = 0.27, 95% CI: 0.12, 0.42; high concentration β = 0.24, 95% CI: 0.09, 0.40). However, the medium concentration of PFOA (β = −0.23, 95% CI: −0.38, −0.08) was correlated with a decreased frequency of common cold (Table 3).

Regarding tonsillitis, the high group of PFDoDA (OR = 1.92, 95% CI: 1.09, 3.38), and both the medium and the high group of PFTrDA (medium concentration OR = 2.22, 95% CI: 1.26, 3.90; high concentration OR = 1.91, 95% CI: 1.07, 3.42) were positively correlated with the incidence of tonsillitis (Table 2). Several PFASs were significantly positively correlated with the frequency of tonsillitis (Table 3). Specifically, medium and high PFNA (medium concentration β = 0.57, 95% CI: 0.22, 0.93; high concentration β = 0.38, 95% CI: 0.00, 0.76), high PFDA (β = 0.45, 95% CI: 0.10, 0.79), high PFUnDA (β = 0.42, 95% CI: 0.08, 0.76), medium and high PFDoDA (medium concentration β = 0.42, 95% CI: 0.02, 0.81; high concentration β = 0.66, 95% CI: 0.29, 1.04), medium and high PFTrDA (medium concentration β = 0.87, 95% CI: 0.47, 1.26; high concentration β = 0.76, 95% CI: 0.36, 1.16), high 6:2 Cl-PFESA (β = 0.37, 95% CI: 0.01, 0.72), and medium and high 8:2 Cl-PFESA (medium concentration β = 0.39, 95% CI: 0.03, 0.75; high concentration β = 0.47, 95% CI: 0.12, 0.83) were all associated with an increase in the frequency of tonsillitis. Nonetheless, medium PFOA concentration was associated with a decrease in the frequency of tonsillitis (β = 0.37, 95% CI: −0.73, 0.00).

In addition, Appendix A showed the association between PFASs exposure, treated as continuous variables, and RTIs in preschool children. PFDoDA is related to an increased odds of tonsillitis (for each doubling of concentration, OR = 1.32; 95% CI: 1.01, 1.71) (Appendix A). Perfluorotridecanoic acid (PFTrDA) is significantly linked to a rise in the frequency of common cold (β = 0.09, 95% CI: 0.00, 0.18). Several PFASs are significantly associated with an increased occurrence of tonsillitis, including perfluorononanoic acid (PFNA) (β = 0.24, 95% CI: 0.01, 0.48), perfluorodecanoic acid (PFDA) (β = 0.21, 95% CI: 0.03, 0.40), perfluoroundecanoic acid (PFUnDA) (β = 0.21, 95% CI: 0.01, 0.42), PFDoDA (β = 0.30, 95% CI: 0.13, 0.47), PFTrDA (β = 0.22, 95% CI: 0.01, 0.43), and 8:2 chlorinated polyfluorinated ether sulfonic acid (8:2 Cl-PFESA) (β = 0.15, 95% CI: 0.01, 0.29) (Appendix A).

This research did not identify any significant associations between PFASs and the incidence or frequency of bronchitis and pneumonia. Additionally, Appendix A displays the results of the sensitivity analysis. After excluding factors such as preterm birth, low birth weight, and macrosomia, the results were generally consistent with those in the total population.

### 3.4. Results of Sex-Stratified Analyses

When stratified by child sex, there were no statistically significant associations between PFASs and the incidence of RTIs (Appendix A). Using Poisson regression analysis, we found that some PFASs were associated with the frequency of RTIs (Appendix A). PFDoDA was significantly associated with increased frequency of tonsillitis (β = 0.25, 95% CI: 0.02, 0.48) in boys. In girls, however, several PFASs showed a significant correlation with an increased frequency of tonsillitis after prenatal exposure, including PFDA (β = 0.32, 95% CI: 0.03, 0.62), PFUnDA (β = 0.41, 95% CI: 0.10, 0.73), and PFDoDA (β = 0.34, 95% CI: 0.09, 0.60).

### 3.5. Associations between PFAS Mixture and RTIs

According to the result of the WQS analysis in Table 4 and Figure 1, after accounting for covariates, a significant association was observed between PFAS mixture and increased frequency of the common cold (β = 0.07; 95% CI: 0.01, 0.13), with PFTrDA contributing the most (41.75%).

For tonsillitis, mixed PFASs showed a significant association with both increased incidence (OR = 1.29, 95% CI: 1.02, 1.62) and frequency of tonsillitis (β = 0.33; 95% CI: 0.12, 0.54). PFDoDA was the dominant contributor to these correlations, accounting for 48.84% and 42.02%, respectively.

After adjusting for covariates, the Quantile g-computation analysis indicated that the weight of the contribution of PFASs mixture in the combined effects was broadly consistent with the results of the WQS regression (Appendix A).

## 4. Discussion

In a prospective birth cohort, our study investigated the association between umbilical cord blood exposures to 10 PFASs, including Cl-PFESAs, and the incidence and frequency of RTIs. Our results confirmed that both single and mixed PFASs were significantly linked to an increased frequency of common colds and an elevated risk of tonsillitis events. Notably, PFDoDA, PFTrDA, PFDA, and 8:2 Cl-PFESA were particularly prominent. When stratified by sex, we observed that certain PFASs were associated with an increase in frequency of tonsillitis in both boys and girls. However, there was no significant interaction between PFASs exposure and sex.

Due to the homologous specificity effect of maternal–fetal transfer, the PFASs levels in umbilical cord blood more accurately reflect the risk of exposure compared to maternal blood. In most studies, PFOS and PFOA were the two highest PFAS in cord serum, but PFNA levels were highest in the Guangzhou birth cohort [21]. In our study, PFOS (4.17 ng/mL) and PFOA (1.62 ng/mL) had lower levels than those in the United States (24.00, 5.60 ng/mL) [22], Canada (4.77, 2.11 ng/mL) [23], and Taiwan, China (5.11, 2.55 ng/mL) [24], but significantly higher than those in South Korea (0.91, 0.66 ng/mL) [25] and Belgium (0.73, 0.68 ng/mL) [26]. The median level of 6:2 Cl-PFESA (0.79 ng/mL) ranked third in the target PFASs, surpassing Maoming (0.32 ng/mL) [27]. The 6:2 Cl-PFESA and 8:2 Cl-PFESA, as alternatives to PFOS and PFOA, were detected to a limited extent in current biometric data, suggesting that the internal burden might be broad in mothers and infants.

Limited epidemiological research connecting PFAS exposures to RTIs in Preschool Children has yielded inconsistent results. In the single-pollutant model, our study found a significant association between exposure to PFTrDA and PFHxS and an increased incidence of common cold in preschool children. Likewise, exposure to PFUnDA, PFDoDA, and PFTrDA resulted in a significant association with an increased frequency of common cold. Similar to our results, many studies have observed positive associations between PFASs exposure and common cold [9,16,17]. In the Norwegian MoBa cohort, a study found higher PFUnDA was positively associated with an increased number of colds in children aged 0–2 years [16], while another study discovered that PFOA and PFNA had the same influence in participants who at age 3 and throughout the ages of 0–3 [9]. The Danish Odense cohort study found that PFOS was significantly associated with more days of fever, and PFOA was related to high risk of concurrent fever and runny nose [17]. While the Laizhou Bay cohort found that exposure to 10 PFASs was associated with an increase in the frequency of common cold, individual PFASs exposure did not demonstrate any statistically significant associations with the frequency of common cold [28]. Additionally, our study found that compared to the low concentration PFOA group, the medium had fewer cold occurrences.

Additionally, our study did not observe a significant association between PFAS exposure and bronchitis or pneumonia. Research in Hokkaido, Japan, revealed a positive correlation between exposure to PFOA, PFDA, and PFDoDA and an increased risk of pneumonia in children up to age 7 [29]. Moreover, findings from the Norwegian MoBa cohort study suggested a higher incidence of bronchitis or pneumonia in children aged 0–3 associated with mid-pregnancy serum levels of PFOA, PFOS, and PFHxS [18].

Considering simultaneous exposure to several PFASs, we employed WQS regression to analyze the relationship between prenatal mixed PFASs exposure and RTIs in preschool children. Overall, mixed PFAS exposure was significantly associated with an increase in the frequency of common cold, and a higher incidence and frequency of tonsillitis. The contributions, PFUnDA, PFDoDA, PFTrDA, and 8:2 Cl-PFESA were particularly notable. To our knowledge, only one study from the Laizhou Bay cohort in Shandong had previously used WQS regression to explore this relationship [28]. This study, based on blood samples collected from expectant mothers, analyzed 10 PFASs and found that mixed PFASs exposure correlated with an increased number of cold and a higher probability of diarrhea occurrence and frequency at 13–14 months of age. Our findings are consistent with this study, collectively suggesting that prenatal mixed PFASs exposure may increase the risk of infectious diseases in children.

Our study differs from prior research due to several factors. Firstly, the levels of PFASs exposure among populations in different studies vary. The median concentration of PFOA in the Norwegian MoBa cohort was 1.1 ng/mL, which is lower than the 1.62 ng/mL observed in our study [16]. In contrast, the median concentration of PFOS detected in the Danish Odense cohort (8.07 ng/mL) is twice that of our study, which stands at 4.17 ng/mL [17]. Secondly, the samples tested for PFASs in each study differ. We measured the concentration in umbilical cord serum, while the Danish Odense cohort tested serum from early pregnancy [17]. Currently, the outcomes of studies on the association between prenatal PFASs exposure and colds are inconsistent, indicating a need for further epidemiological research to delve deeper and confirm these associations.

In both single and combined exposure models, long-chain PFASs, with a higher number of carbon atoms, had a greater contribution to the risk of RTIs in preschool children. Many studies have shown that the longer the carbon chain of PFASs, the greater their binding affinity to human serum proteins, leading to a greater likelihood of bioaccumulation [30,31]. In vitro experiments have found that PFASs with more carbon atoms have a lower IC50 value, suggesting that long-chain PFASs may be harmful to organisms [32]. Previous research has mainly focused on PFASs such as PFOA and PFOS, which have higher concentrations in environmental media and in organisms. More attention should be paid to longer chain PFASs such as PFDoDA, PFUnDA, and PFHxS.

Owing to the extensive evidence linking PFOS exposure to harm in the population, it was banned. In its place, Cl-PFESAs emerged as a widely used substitute for PFOS [33]. The main component of Cl-PFESAs is 6:2 Cl-PFESA, while 8:2 Cl-PFESA is an impurity produced during the manufacturing process [34]. In our study, we observed that the concentration of 6:2 Cl-PFESA (0.79 ng/mL) was several magnitudes higher than that of 8:2 Cl-PFESA (0.03 ng/mL), consistent with their concentration ratios found in the environment and other biological samples [35]. Our study also confirmed that the detection rates for both 8:2 Cl-PFESA and 6:2 Cl-PFESA exceed 95%, suggesting widespread exposure to Cl-PFESAs among newborns in Wuhan. Both 6:2 Cl-PFESA and 8:2 Cl-PFESA showed a significant association with an increased incidence of tonsillitis. Although the concentration of 8:2 Cl-PFESA is relatively lower than that of 6:2 Cl-PFESA, its impact appears more pronounced. At the same time, the WQS model highlighted that 8:2 Cl-PFESA contributed to over a fifth of the positive association between PFASs mixtures and tonsillitis occurrences.

Current evidence suggests that the toxic effects of 8:2 Cl-PFESA may be stronger than those of PFOS and 6:2 Cl-PFESA. In a zebrafish experiment, compared to PFOS and 6:2 Cl-PFESA, 8:2 Cl-PFESA induced liver toxicity via different lipid regulatory mechanisms and had a markedly increased antagonistic effect on peroxisome proliferator-activated receptors [36]. What is more concerning is that the human half-life of Cl-PFESAs (6:2 Cl-PFESA: 6.2 years) exceeded that of conventional PFASs (PFOS: 6.7 years). Additionally, Cl-PFESAs (6:2 Cl-PFESA: 40.3%; 8:2 CI-PFESA: 55.7%) has a higher placental transfer rate compared to PFOS (40.3%) [37]. Simultaneously, 8:2 Cl-PFESA has a lower renal clearance rate, making it harder to be excreted from the body [38]. Previous research has also linked Cl-PFESAs to adverse birth outcomes, such as premature birth and low birth weight, as well as changes in vitamin D and hormone levels [20,39,40,41]. Epidemiological evidence also suggested that Cl-PFESAs further affected adult health, including hypertension, thyroid hormone disturbances, dyslipidemia, and liver function abnormalities [42,43,44,45]. Thus, potential health risks should be considered when using these novel PFASs as substitutes.

When interpreting the results of this study, it is important to acknowledge several limitations. First, the study used PFAS concentrations in umbilical cord blood as an indicator. However, breastfeeding is another important route for newborns to be exposed to PFASs. Second, the lack of data limits our full understanding of its impact and related biological mechanisms. Furthermore, further research is imperative for a comprehensive understanding. Third, the outcomes of RTIs in children and information on covariates were obtained through standardized questionnaires completed by the children’s parents, which may introduce bias that could affect the study results. Finally, although we adjusted for many items, other potential confounding factors like dietary intake were not considered.

In summary, this study provides relatively reliable epidemiological evidence for the association between prenatal PFASs exposure and RTIs in preschool children. It is the first to examine the impact of the PFOS substitute, Cl-PFESAs. Additionally, the findings from our sensitivity analysis reveal that our study is robust because our results mirror those of the total population, even after the exclusion of some variables.

## 5. Conclusions

Our study found a widespread exposure to PFASs among pregnant women in Wuhan. The results suggest that both single and mixed exposures to different PFASs, including the newer alternatives (6:2 Cl-PFESA, 8:2 Cl-PFESA), correlate with an increased incidence and frequency of diseases such as the common cold and tonsillitis. These data imply that PFASs exposure may increase risk in offspring, although we did not find sex-specific variations. Based on these findings, it is imperative for future research and relevant agencies to increase their focus on the potential health effects of PFASs on children.

## Figures and Tables

**Figure 1 toxics-11-00897-f001:**
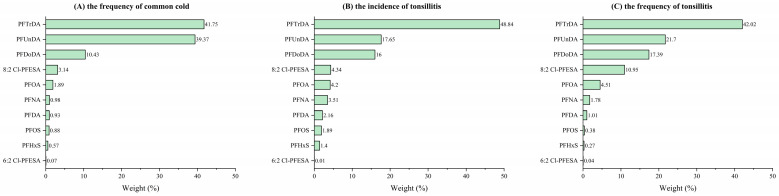
Weights of each single PFAS contributing to the association between PFAS mixture and RTIs. The figure shows the results of WQS regression.

**Table 1 toxics-11-00897-t001:** Distribution of umbilical cord serum PFAS concentrations (ng/mL).

Compounds	LOQ	DR (%)	GM	Selected Percentiles				
				P5	P25	P50	P75	P95
PFOA	0.02	100.00	1.63	0.96	1.29	1.62	1.98	3.11
PFNA	0.02	100.00	0.28	0.13	0.21	0.28	0.37	0.58
PFDA	0.02	100.00	0.16	0.07	0.11	0.16	0.22	0.38
PFUnDA	0.01	100.00	0.19	0.09	0.14	0.20	0.27	0.43
PFDoDA	0.01	90.89	0.02	<LOQ	0.02	0.03	0.04	0.06
PFTrDA	0.01	99.81	0.11	0.05	0.08	0.11	0.15	0.22
PFHxS	0.02	100.00	0.43	0.20	0.31	0.42	0.57	0.91
PFOS	0.01	100.00	4.40	1.43	2.77	4.17	6.47	13.84
6:2 Cl-PFESA	0.01	100.00	0.79	0.30	0.57	0.79	1.13	2.27
8:2 Cl-PFESA	0.01	95.83	0.03	<LOQ	0.02	0.03	0.04	0.10

Abbreviations: LOQ, limit of quantification; DR, detection rate; GM, geometric mean.

**Table 2 toxics-11-00897-t002:** Association between single PFASs (categorical variables) and the incidence of RTIs.

	Low Concentration	Medium Concentration		High Concentration		*P* _trend_
		OR (95% CI)	*p*	OR (95% CI)	*p*	
PFOA (ng/mL)	<1.40	1.40–1.86		>1.86		
Common cold	Ref	0.81 (0.4, 1.37)	0.408	0.95 (0.56, 1.63)	0.805	0.916
Bronchitis	Ref	0.80 (0.51, 1.25)	0.531	0.81 (0.52, 1.28)	0.649	0.410
Pneumonia	Ref	1.29 (0.67, 2.49)	0.465	1.10 (0.56, 2.15)	0.912	0.842
Tonsillitis	Ref	0.64 (0.37, 1.11)	0.117	0.90 (0.53, 1.51)	0.627	0.781
PFNA (ng/mL)	<0.23	0.23–0.33		>0.33		
Common cold	Ref	0.89 (0.52, 1.51)	0.986	0.79 (0.46, 1.35)	0.456	0.395
Bronchitis	Ref	1.50 (0.96, 2.37)	0.114	1.21 (0.76, 1.93)	0.961	0.466
Pneumonia	Ref	1.29 (0.60, 2.42)	0.323	0.95 (0.48, 1.89)	0.559	0.869
Tonsillitis	Ref	1.38 (0.81, 2.38)	0.424	1.32 (0.76, 2.30)	0.625	0.352
PFDA (ng/mL)	<0.13	0.13–0.20		>0.20		
Common cold	Ref	0.93 (0.55, 1.58)	0.996	0.87 (0.51, 1.47)	0.637	0.598
Bronchitis	Ref	1.33 (0.85, 2.08)	0.309	1.18 (0.75, 1.88)	0.891	0.571
Pneumonia	Ref	1.52 (0.80, 2.89)	0.207	1.15(0.58, 2.30)	0.819	0.809
Tonsillitis	Ref	1.19 (0.68, 2.06)	0.863	1.53 (0.89, 2.63)	0.147	0.121
PFUnDA (ng/mL)	<0.16	0.16–0.24		>0.24		
Common cold	Ref	0.85 (0.50, 1.44)	0.568	0.93 (0.55, 1.60)	0.949	0.823
Bronchitis	Ref	1.47 (0.93, 2.31)	0.151	1.22 (0.77, 1.93)	0.967	0.424
Pneumonia	Ref	1.52 (0.78, 2.95)	0.376	1.40 (0.71, 2.76)	0.659	0.345
Tonsillitis	Ref	1.07 (0.61, 1.87)	0.477	1.61 (0.94, 2.75)	0.055	0.073
PFDoDA (ng/mL)	<0.02	0.02–0.03		>0.03		
Common cold	Ref	1.67 (0.98, 2.87)	0.113	1.30 (0.78, 2.17)	0.988	0.402
Bronchitis	Ref	1.23 (0.78, 1.94)	0.502	1.15 (0.73, 1.83)	0.835	0.591
Pneumonia	Ref	1.67 (0.86, 3.25)	0.157	1.26 (0.63, 2.51)	0.927	0.629
Tonsillitis	Ref	1.52 (0.85, 2.71)	0.708	1.92 (1.09, 3.38)	0.021 *	0.025 *
PFTrDA (ng/mL)	<0.09	0.09–0.14		>0.14		
Common cold	Ref	2.00 (1.18, 3.41)	0.035 *	1.43 (0.86, 2.39)	0.956	0.230
Bronchitis	Ref	1.02 (0.65, 1.60)	0.512	1.35 (0.85, 2.12)	0.147	0.176
Pneumonia	Ref	1.26 (0.65, 2.43)	0.732	1.30 (0.67, 2.53)	0.605	0.468
Tonsillitis	Ref	2.22 (1.26, 3.90)	0.040 *	1.91 (1.07, 3.42)	0.023 *	0.064
PFHxS (ng/mL)	<0.34	0.34–0.51		>0.51		
Common cold	Ref	1.68 (1.00, 2.91)	0.030 *	0.97 (0.58, 1.60)	0.202	0.706
Bronchitis	Ref	1.18 (0.75, 1.85)	0.428	1.02 (0.65, 1.61)	0.757	0.999
Pneumonia	Ref	1.27 (0.66, 2.45)	0.588	1.20 (0.61, 2.34)	0.843	0.649
Tonsillitis	Ref	0.98 (0.57, 1.69)	0.706	1.15 (0.68, 1.96)	0.520	0.575
PFOS (ng/mL)	<3.48	3.48–5.49		>5.49		
Common cold	Ref	0.79 (0.46, 1.34)	0.660	0.75 (0.44, 1.29)	0.485	0.359
Bronchitis	Ref	0.85 (0.54, 1.33)	0.460	0.96 (0.61, 1.51)	0.832	0.977
Pneumonia	Ref	0.79 (0.40, 1.57)	0.173	1.43 (0.76, 2.73)	0.093	0.168
Tonsillitis	Ref	1.13 (0.67, 1.89)	0.301	0.79 (0.45, 1.37)	0.228	0.317
6:2 Cl-PFESA (ng/mL)	<0.63	0.63–0.97		>0.97		
Common cold	Ref	1.07 (0.63, 1.82)	0.660	0.94 (0.55, 1.60)	0.674	0.781
Bronchitis	Ref	0.92 (0.58, 1.45)	0.394	1.19 (0.75, 1.88)	0.287	0.408
Pneumonia	Ref	1.06 (0.56, 2.01)	0.879	1.02 (0.53, 1.99)	0.990	0.953
Tonsillitis	Ref	1.13 (0.66, 1.94)	0.968	1.25 (0.72, 2.16)	0.497	0.433
8:2 Cl-PFESA (ng/mL)	<0.02	0.02–0.04		>0.04		
Common cold	Ref	0.96 (0.57, 1.63)	0.943	0.90 (0.53, 1.52)	0.699	0.682
Bronchitis	Ref	1.30 (0.83, 2.04)	0.427	1.23 (0.78, 1.93)	0.703	0.468
Pneumonia	Ref	1.40 (0.73, 2.69)	0.499	1.33 (0.69, 2.59)	0.674	0.477
Tonsillitis	Ref	1.28 (0.74, 2.19)	0.572	1.25 (0.73, 2.15)	0.671	0.500

Abbreviation: OR, odds ratio; CI, confidence interval; Ref, Reference. The figure shows the results of Logistic regression after correction for covariates. The model was adjusted for maternal age (continuous variable), family annual income (<CNY 100,000; ≥CNY 100,000), child sex (male; female), gestational week (continuous variable), parity (primipara; multipara), breastfeeding duration (<6 months; ≥6 months), and child exposure to passive smoking (yes; no). These potential confounders were adjusted for in the regression model. *: *p* < 0.05.

**Table 3 toxics-11-00897-t003:** Association between single PFASs (categorical variables) and the frequency of RTIs.

	Low Concentration	Medium Concentration		High Concentration		*P* _trend_
		β (95% CI)	*p*	β (95% CI)	*p*	
PFOA (ng/mL)	<1.40	1.40–1.86		>1.86		
Common cold	Ref	−0.23 (−0.38, −0.08)	0.003 *	−0.03 (−0.18, 0.11)	0.638	0.875
Bronchitis	Ref	−0.22 (−0.48, 0.04)	0.102	−0.04 (−0.29, 0.20)	0.735	0.860
Pneumonia	Ref	0.06 (−0.49, 0.62)	0.822	−0.23 (−0.82, 0.37)	0.454	0.431
Tonsillitis	Ref	−0.37 (−0.73, 0.00)	0.048 *	−0.02 (−0.34, 0.31)	0.910	0.943
PFNA (ng/mL)	<0.23	0.23–0.33		>0.33		
Common cold	Ref	−0.07 (−0.22, 0.07)	0.326	0.04 (−0.11, 0.19)	0.595	0.535
Bronchitis	Ref	0.09 (−0.16, 0.34)	0.465	0.06 (−0.20, 0.32)	0.652	0.668
Pneumonia	Ref	0.08 (−0.46, 0.61)	0.778	−0.40 (−1.03, 0.23)	0.211	0.224
Tonsillitis	Ref	0.57 (0.22, 0.93)	0.002 *	0.38 (0.00, 0.76)	0.049 *	0.082
PFDA (ng/mL)	<0.13	0.13–0.20		>0.20		
Common cold	Ref	−0.03 (−0.18, 0.12)	0.712	0.10 (−0.04, 0.25)	0.173	0.122
Bronchitis	Ref	0.00 (−0.25, 0.25)	0.983	0.05 (−0.21, 0.30)	0.714	0.695
Pneumonia	Ref	0.15 (−0.39, 0.69)	0.583	−0.19 (−0.81, 0.42)	0.535	0.487
Tonsillitis	Ref	0.05 (−0.32, 0.42)	0.792	0.45 (0.10, 0.79)	0.011 *	0.006 *
PFUnDA (ng/mL)	<0.16	0.16–0.24		>0.24		
Common cold	Ref	0.05 (−0.10, 0.20)	0.526	0.16 (0.01, 0.31)	0.036 *	0.034 *
Bronchitis	Ref	0.16 (−0.09, 0.41)	0.211	0.12 (−0.14, 0.37)	0.384	0.395
Pneumonia	Ref	0.33 (−0.23, 0.89)	0.252	0.00 (−0.61, 0.61)	0.999	0.996
Tonsillitis	Ref	−0.05 (−0.43, 0.32)	0.784	0.42 (0.08, 0.76)	0.015 *	0.010 *
PFDoDA (ng/mL)	<0.02	0.02–0.03		>0.03		
Common cold	Ref	0.29 (0.14, 0.45)	<0.001 *	0.22 (0.06, 037)	0.006 *	0.019 *
Bronchitis	Ref	0.23 (−0.02, 0.49)	0.075	0.07 (−0.19, 0.34)	0.596	0.732
Pneumonia	Ref	0.37 (−0.22, 0.96)	0.221	0.20 (−0.41, 0.81)	0.512	0.604
Tonsillitis	Ref	0.42 (0.02, 0.81)	0.037 *	0.66 (0.29, 1.04)	<0.001 *	0.001 *
PFTrDA (ng/mL)	<0.09	0.09–0.14		>0.14		
Common cold	Ref	0.27 (0.12, 0.42)	<0.001 *	0.24 (0.09, 0.40)	0.002 *	0.007 *
Bronchitis	Ref	−0.04 (−0.30, 0.22)	0.772	0.17 (−0.09, 0.42)	0.199	0.157
Pneumonia	Ref	0.19 (−0.38, 0.75)	0.522	0.07 (−0.53, 0.67)	0.818	0.874
Tonsillitis	Ref	0.87 (0.47, 1.26)	<0.001 *	0.76 (0.36, 1.16)	<0.001 *	0.002 *
PFHxS (ng/mL)	<0.34	0.34–0.51		>0.51		
Common cold	Ref	−0.05 (−0.19, 0.10)	0.518	−0.11 (−0.25, 0.04)	0.154	0.155
Bronchitis	Ref	−0.02 (−0.26, 0.23)	0.884	−0.21 (−0.47, 0.05)	0.121	0.112
Pneumonia	Ref	0.22 (−0.37, 0.82)	0.461	0.33 (−0.26, 0.92)	0.269	0.282
Tonsillitis	Ref	0.01 (−0.33, 0.36)	0.932	−0.07 (−0.41, 0.27)	0.688	0.667
PFOS (ng/mL)	<3.48	3.48–5.49		>5.49		
Common cold	Ref	−0.07 (−0.22, 0.08)	0.348	−0.08 (−0.22, 0.07)	0.313	0.372
Bronchitis	Ref	−0.19 (−0.43, 0.06)	0.143	−0.15 (−0.40, 0.11)	0.260	0.342
Pneumonia	Ref	−0.47 (−1.08, 0.14)	0.134	0.09 (−0.46, 0.64)	0.744	0.498
Tonsillitis	Ref	−0.04 (−0.37, 0.28)	0.790	−0.35 (−0.71, 0.01)	0.059	0.051
6:2 Cl-PFESA (ng/mL)	<0.63	0.63–0.97		>0.97		
Common cold	Ref	−0.03 (−0.18, 0.12)	0.666	0.00 (−0.15, 0.15)	0.991	0.946
Bronchitis	Ref	−0.10 (−0.36, 0.16)	0.448	0.10 (−0.15, 0.35)	0.436	0.376
Pneumonia	Ref	−0.09 (−0.65, 0.46)	0.739	−0.18 (−0.77, 0.41)	0.550	0.552
Tonsillitis	Ref	0.19 (−0.17, 0.55)	0.305	0.37 (0.01, 0.72)	0.042 *	0.042 *
8:2 Cl-PFESA (ng/mL)	<0.02	0.02–0.04		>0.04		
Common cold	Ref	0.12 (−0.02, 0.27)	0.095	0.04 (−0.11, 0.19)	0.643	0.891
Bronchitis	Ref	0.19 (−0.06, 0.44)	0.141	0.06 (−0.19, 0.32)	0.624	0.806
Pneumonia	Ref	−0.04 (−0.60, 0.51)	0.874	−0.18 (−0.77, 0.40)	0.535	0.526
Tonsillitis	Ref	0.39 (0.03, 0.75)	0.034 *	0.47 (0.12, 0.83)	0.010 *	0.019 *

Abbreviation: OR, odds ratio; CI, confidence interval; Ref, Reference. The figure shows the results of Poisson regression after correction for covariates. The model was adjusted for maternal age (continuous variable), family annual income (<CNY 100,000; ≥CNY 100,000), child sex (male; female), gestational week (continuous variable), parity (primipara; multipara), breastfeeding duration (<6 months; ≥6 months), and child exposure to passive smoking (yes; no). These potential confounders were adjusted for in the regression model. *: *p* < 0.05.

**Table 4 toxics-11-00897-t004:** Association between PFAS mixture and RTIs.

	Positive Association		Negative Association	
	OR (95% CI)	*p*	OR (95% CI)	*p*
Incidence				
Common cold	1.12 (0.87, 1.43)	0.386	0.93 (0.72, 1.19)	0.553
Bronchitis	1.09 (0.87, 1.35)	0.455	0.96 (0.77, 1.21)	0.749
Pneumonia	1.20 (0.86, 1.67)	0.286	1.05(0.76, 1.45)	0.770
Tonsillitis	1.29 (1.02, 1.62)	0.033 *	0.96(0.74, 1.24)	0.760
Frequency				
Common cold	0.07 (0.01, 0.13)	0.024 *	−0.04 (−0.11, 0.03)	0.310
Bronchitis	0.01 (−0.10, 0.13)	0.838	−0.08 (−0.21, 0.04)	0.189
Pneumonia	0.18 (−0.09, 0.45)	0.185	−0.19 (−0.47, 0.10)	0.195
Tonsillitis	0.33 (0.12, 0.54)	0.002 *	−0.08 (−0.24, 0.09)	0.367

Abbreviation: OR, odds ratio; CI, confidence interval. The model was adjusted for maternal age (continuous variable), family annual income (<CNY 100,000; ≥CNY 100,000), child sex (male; female), gestational week (continuous variable), parity (primipara; multipara), breastfeeding duration (<6 months; ≥6 months), and child exposure to passive smoking (yes; no). These potential confounders were adjusted for in the regression model. *: *p* < 0.05.

## Data Availability

Not applicable.

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
