# Peer review of "The Association between Prenatal Exposure to Per- and Polyfluoroalkyl Substances and Respiratory Tract Infections in Preschool Children: A Wuhan Cohort Study"

_toxics, 2023, doi:10.3390/toxics11110897_

Round 1
Reviewer 1 Report
Comments and Suggestions for Authors
The effects of PFASs on Respiratory tract infections were studied. Sufficient numbers of cases were investigated in this study. This study is important from the public health perspectives. Methods and analysis is reasonable. Although some P-values associated with PFASs and Respiratory tract infections are large, this trial has scientific meaning.
Author Response
Response to Reviewer 1 Comments
Point: The effects of PFASs on Respiratory tract infections were studied. Sufficient numbers of cases were investigated in this study. This study is important from the public health perspectives. Methods and analysis is reasonable. Although some P-values associated with PFASs and Respiratory tract infections are large, this trial has scientific meaning.
Response: Thank you for your appreciation of our research. Your valuable comments and insights have been helped to improve the quality and impact of our study. Firstly, we are pleased to note that you acknowledge the sufficient number of cases we investigated, with 527 mother-infant pairs included from the Wuhan Healthy Baby Cohort (WHBC) in China. Adequate sample size is essential to ensure the robustness of our findings. Secondly, previous studies on the association between PFASs and RTIs shows inconsistency. Our aim is to enhance comprehension of the relationship, and we appreciate your recognition of the significance of our research in this area. Lastly, We acknowledge your concern about the large P-values. It's important to emphasize that while some P-values may seem large, their significance should be interpreted within the context of the overall investigation. We are mindful of the complex nature of the relationship between PFASs and RTIs, and we believe that our study has scientific meaning and implications beyond individual P-values.

Reviewer 2 Report
Comments and Suggestions for Authors
The epidemiological study presented in this manuscript presents interesting results. We have to deal with the fact that we are exposed to these volatile and persistent contaminants that we now find everywhere. It is important to understand what consequences are caused by chronic exposure. In studies of this type, the relationship between the effect and exposure to a single contaminant is obviously missing, which of the many is most harmful or what specific effects each of them could have.
Author Response
Response to Reviewer 2 Comments
Point 1: The epidemiological study presented in this manuscript presents interesting results. We have to deal with the fact that we are exposed to these volatile and persistent contaminants that we now find everywhere. It is important to understand what consequences are caused by chronic exposure.
Response 1: We appreciate your review, and we are pleased to found our results interesting. We understand and share your concern about the widespread exposure to volatile and persistent contaminants, such as PFAS, in our environment. Our study aims to illuminate the potential health effects of prenatal exposure to PFASs and the incidence and frequency of RTIs in preschool children. Your feedback reinforces the importance of our research, and we are dedicated to enhancing our understanding of the consequences of chronic exposure to pollutants like PFAS. We will persist in refining our work to provide valuable perspectives on this critical public health issue.
Point 2: In studies of this type, the relationship between the effect and exposure to a single contaminant is obviously missing, which of the many is most harmful or what specific effects each of them could have.
Response 2: Thanks for the suggestion. According to your suggestion, we have included the association between exposure to a single PFSAs and RTIs in the Discussion section (Lines 273-295, page 9-10).
“Limited epidemiological research connecting PFASs exposures to RTIs in Preschool Children has yielded inconsistent results. In the single-pollutant model, our study found a significant association between exposure to PFTrDA and PFHxS and an in-creased incidence of common cold in preschool children. Likewise, exposure to PFUnDA, PFDoDA, and PFTrDA resulted in a significant association with an increased frequency of common cold. Similar to our results, many studies have observed positive associations between PFASs exposure and common cold [9,16,17]. In the Norwegian MoBa cohort, a study found higher PFUnDA was positively associated with an in-creased number of colds in children aged 0-2 years [16], while another study discovered that PFOA and PFNA had the same influence in participants who at age 3 and throughout the ages of 0-3 [9]. The Danish Odense cohort study found that PFOS was significantly associated with more days of fever, and PFOA was related to high risk of concurrent fever and runny nose [17]. While the Laizhou Bay cohort found that exposure to 10 PFASs was associated with an increase in the frequency of common cold, indi-vidual PFASs exposure did not demonstrate any statistically significant associations with the frequency of common cold [28]. Additionally, our study found that compared to the low concentration PFOA group, the medium had fewer cold occurrences.
Besides, our study did not observe a significant association between PFAS exposure and bronchitis or pneumonia. Research in Hokkaido, Japan, revealed a positive corre-lation between exposure to PFOA, PFDA, and PFDoDA and an increased risk of pneumonia in children up to age 7 [29]. Moreover, findings from the Norwegian MoBa cohort study suggested a higher incidence of bronchitis or pneumonia in children aged 0-3 associated with mid-pregnancy serum levels of PFOA, PFOS, and PFHxS [18].”

Reviewer 3 Report
Comments and Suggestions for Authors
The paper has some serious errors. So it cannot be published.
The presentation is poor. There are erroneous expressions. The presentation of the results is poor. There are too many tables and too much data. The results presented in tables are repeated in the text. The number of tables and the number of figures must be simplified.
Introduction section is very poor. The authors focus mainly on two studies (Impinen et al. and Lebeaux et al.) however, there are numerous papers that have focused on the effects of PFASs exposure, and the conclusions are conflicting. In this regard, the authors comment that Impinen et al. studies a Danish birth cohort. However, these authors studied two subcohorts of the Norwegian Mother and Child Cohort Study.
They later cite Lebeaux et al. in relation to a Norwegian birth cohort. It is a new error because these authors study a prospective cohort in the greater Cincinnati, Ohio region.
There are also important studies that are not commented on by the authors (Okada et al. 2012; Granum et al. 2013; Impinen et al. 2018; Dalsager et al. 2016; Goudarzi et al. 2017, etc.).
Other errors:
Line 123. Interaction.
Line 147. Error: Not 7.82% but 47.82.
Minor recommendations are the following:
The objectives of the study should be reviewed and the primary and secondary objectives should be specified.
The acronyms used must be explained both in the abstract section and in the text.
Line 144. The average age of pregnant women must be specified.
Lines 230 to 232: This expression must be explained in the discussion section.
Comments on the Quality of English LanguageMinor editing of English language required
Author Response
Response to Reviewer 3 Comments
Point 1: The presentation is poor. There are erroneous expressions. The presentation of the results is poor. There are too many tables and too much data. The results presented in tables are repeated in the text. The number of tables and the number of figures must be simplified.
Response 1: We thank you for reviewing our manuscript. In response to your valuable feedback, we have revised the text to improve clarity and readability.
- Presentation and Erroneous Expressions:
We recognize the issues with presentation and errors in the original manuscript. We have added abbreviations to improve the clarity and accuracy of our work.
- Excessive Tables and Data:
We appreciate your feedback on the excessive number of tables and data. In the original manuscript we examined the effects of PFASs as continuous variables and as categorical variables on the incidence and frequency of RTIs. This dual presentation may seem redundant. To address this, we have simplified the presentation of results and reduced the number of figures. As a result, we have moved the figures showing PFASs as continuous variables from the original article to the Supplementary Material.
“Figure 1. Association between single PFASs and the incidence of RTIs (PFASs as continuous variables).
Figure 2. Association between single PFASs and the frequency of RTIs (PFASs as continuous variables).”
- Repetition of Results:
To avoid duplication of results in the text and tables, we revised the presentation by removing “Table 1. General demographic characteristics of pregnant women included in the study.” to the Supplementary Material. This streamlining makes the manuscript clearer and more readable.
Point 2: Introduction section is very poor. The authors focus mainly on two studies (Impinen et al. and Lebeaux et al.) however, there are numerous papers that have focused on the effects of PFASs exposure, and the conclusions are conflicting. In this regard, the authors comment that Impinen et al. studies a Danish birth cohort. However, these authors studied two subcohorts of the Norwegian Mother and Child Cohort Study.
Response 2: We appreciate the reviewer for identifying the issue. According to your suggestion, we have rectified the relevant references and conducted a thorough review of all cross-references in the article.
“For example, a Danish birth cohort showed that PFOS and PFOA exposure correlated with an increased risk of hospitalization for lower RTIs in children [15]. “
- Dalsager, L.; Christensen, N.; Halekoh, U.; Timmermann, C.A.G.; Nielsen, F.; Kyhl, H.B.; Husby, S.; Grandjean, P.; Jensen, T.K.; Andersen, H.R. Exposure to perfluoroalkyl substances during fetal life and hospitalization for infectious disease in childhood: A study among 1,503 children from the Odense Child Cohort. Environment International 2021, 149, 106395, doi:10.1016/j.envint.2021.106395.
Point 3: They later cite Lebeaux et al. in relation to a Norwegian birth cohort. It is a new error because these authors study a prospective cohort in the greater Cincinnati, Ohio region.
Response 3: We are grateful for the suggestion. Following your guidance, we have corrected the relevant references and carefully checked the cross-references of all references in the article (Lines 59-62, page 2).
“Conversely, a Norwegian birth cohort found a negative association between PFOS and PFOA exposure and the risk of common colds in children aged 0-3 years, while perfluorohexane sulfonate (PFHxS) exposure was correlated with a surge in the risk of bronchitis, pneumonia, and throat infection [18]”
- Impinen, A.; Longnecker, M.P.; Nygaard, U.C.; London, S.J.; Ferguson, K.K.; Haug, L.S.; Granum, B. Maternal levels of perfluoroalkyl substances (PFASs) during pregnancy and childhood allergy and asthma related outcomes and infections in the Norwegian Mother and Child (MoBa) cohort. Environment international 2019, 124, 462-472, doi:10.1016/j.envint.2018.12.041.
Point 4: There are also important studies that are not commented on by the authors (Okada et al. 2012; Granum et al. 2013; Impinen et al. 2018; Dalsager et al. 2016; Goudarzi et al. 2017, etc.).
Response 4: We genuinely appreciate your valuable suggestion. According to your suggestion, including Point2 and Point3, we have reviewed the references you provided and revised this section to include the relevant studies (Okada et al. 2012; Granum et al. 2013; Impinen et al. 2018; Dalsager et al. 2016) into our manuscript. However, despite our efforts, we could not find this article (Goudarzi et al. 2017). If possible, could you please provide more detailed information or verify the citation? In addition, we have included other important studies in our article to ensure a more comprehensive and robust discussion of the relevant literature.
- Introduction section (Lines 57-64, page 2).
“The effects of PFASs exposure on RTIs in children have been inconsistent. Some studies have reported a positive relationship between PFASs exposure and RTIs [9,15-17]. Conversely, a Norwegian birth cohort found a negative association between PFOS and PFOA exposure and the risk of common colds in children aged 0-3 years, while perfluorohexane sulfonate (PFHxS) exposure was correlated with a surge in the risk of bronchitis, pneumonia, and throat infection [18]. However, findings from a Danish birth cohort revealed no association between PFOS and PFOA and bronchitis, pneumonia and other infections at 18 months of age [19].”
- Discussion section (Lines 273-295, page 9-10).
“Limited epidemiological research connecting PFASs exposures to RTIs in Preschool Children has yielded inconsistent results. In the single-pollutant model, our study found a significant association between exposure to PFTrDA and PFHxS and an in-creased incidence of common cold in preschool children. Likewise, exposure to PFUnDA, PFDoDA, and PFTrDA resulted in a significant association with an increased frequency of common cold. Similar to our results, many studies have observed positive associations between PFASs exposure and common cold [9,16,17]. In the Norwegian MoBa cohort, a study found higher PFUnDA was positively associated with an in-creased number of colds in children aged 0-2 years [16], while another study discovered that PFOA and PFNA had the same influence in participants who at age 3 and throughout the ages of 0-3 [9]. The Danish Odense cohort study found that PFOS was significantly associated with more days of fever, and PFOA was related to high risk of concurrent fever and runny nose [17]. While the Laizhou Bay cohort found that exposure to 10 PFASs was associated with an increase in the frequency of common cold, indi-vidual PFASs exposure did not demonstrate any statistically significant associations with the frequency of common cold [28]. Additionally, our study found that compared to the low concentration PFOA group, the medium had fewer cold occurrences.
Besides, our study did not observe a significant association between PFAS exposure and bronchitis or pneumonia. Research in Hokkaido, Japan, revealed a positive corre-lation between exposure to PFOA, PFDA, and PFDoDA and an increased risk of pneumonia in children up to age 7 [29]. Moreover, findings from the Norwegian MoBa cohort study suggested a higher incidence of bronchitis or pneumonia in children aged 0-3 associated with mid-pregnancy serum levels of PFOA, PFOS, and PFHxS [18].”
- Granum, B.; Haug, L.S.; Namork, E.; Stølevik, S.B.; Thomsen, C.; Aaberge, I.S.; van Loveren, H.; Løvik, M.; Nygaard, U.C. Pre-natal exposure to perfluoroalkyl substances may be associated with altered vaccine antibody levels and immune-related health outcomes in early childhood. Journal of immunotoxicology 2013, 10, 373-379, doi:10.3109/1547691X.2012.755580.
- Dalsager, L.; Christensen, N.; Halekoh, U.; Timmermann, C.A.G.; Nielsen, F.; Kyhl, H.B.; Husby, S.; Grandjean, P.; Jensen, T.K.; Andersen, H.R. Exposure to perfluoroalkyl substances during fetal life and hospitalization for infectious disease in childhood: A study among 1,503 children from the Odense Child Cohort. Environment International 2021, 149, 106395, doi:10.1016/j.envint.2021.106395.
- Impinen, A.; Nygaard, U.; Carlsen, K.L.; Mowinckel, P.; Carlsen, K.; Haug, L.; Granum, B. Prenatal exposure to perfluoralkyl substances (PFASs) associated with respiratory tract infections but not allergy-and asthma-related health outcomes in childhood. Environmental research 2018, 160, 518-523, doi:10.1016/j.envres.2017.10.012.
- Dalsager, L.; Christensen, N.; Husby, S.; Kyhl, H.; Nielsen, F.; Høst, A.; Grandjean, P.; Jensen, T.K. Association between prenatal exposure to perfluorinated compounds and symptoms of infections at age 1–4 years among 359 children in the Odense Child Cohort. Environment international 2016, 96, 58-64, doi:10.1016/j.envint.2016.08.026.
- Impinen, A.; Longnecker, M.P.; Nygaard, U.C.; London, S.J.; Ferguson, K.K.; Haug, L.S.; Granum, B. Maternal levels of perfluoroalkyl substances (PFASs) during pregnancy and childhood allergy and asthma related outcomes and infections in the Norwegian Mother and Child (MoBa) cohort. Environment international 2019, 124, 462-472, doi:10.1016/j.envint.2018.12.041.
- Okada, E.; Sasaki, S.; Saijo, Y.; Washino, N.; Miyashita, C.; Kobayashi, S.; Konishi, K.; Ito, Y.M.; Ito, R.; Nakata, A. Prenatal exposure to perfluorinated chemicals and relationship with allergies and infectious diseases in infants. Environmental research 2012, 112, 118-125, doi:10.1016/j.envres.2011.10.003.
- Wang, Z.; Shi, R.; Ding, G.; Yao, Q.; Pan, C.; Gao, Y.; Tian, Y. Association between maternal serum concentration of perfluoroalkyl substances (PFASs) at delivery and acute infectious diseases in infancy. Chemosphere 2022, 289, 133235, doi:10.1016/j.chemosphere.2021.133235.
- Bamai, Y.A.; Goudarzi, H.; Araki, A.; Okada, E.; Kashino, I.; Miyashita, C.; Kishi, R. Effect of prenatal exposure to per-and polyfluoroalkyl substances on childhood allergies and common infectious diseases in children up to age 7 years: The Hokkaido study on environment and children's health. Environment International 2020, 143, 105979, doi:10.1016/j.envint.2020.105979.
Other errors:
Point 5:
Line 123. Interaction.
Line 147. Error: Not 7.82% but 47.82.
Response 5: We appreciate Reviewer for carefully reviewing our paper. According to your suggestion, we have made the necessary corrections and conducted thorough error-checking.
Line126. “interaction” was deleted.
Line 150.”7.82%” were corrected as “47.82”.
Minor recommendations are the following:
Point 6: The objectives of the study should be reviewed and the primary and secondary objectives should be specified.
Response 6: Thank you for your input. According to your suggestion, we have amended the abstract to clarify the objectives of the study in the abstract section.
“Furthermore, stratified and interaction analyses were performed to evaluate if there were sex-specific associations.” (Lines 23-24, page 1).
Our primary objective is to investigate the association between prenatal exposure to PFASs and the incidence and frequency of RTIs in preschool children. Additionally, our secondary objective is to evaluate sex-specific associations using stratified and interaction analysis.
Point 7: The acronyms used must be explained both in the abstract section and in the text.
Response 7: Thank you for your suggestion. According to your suggestion, we have reviewed and fixed all acronyms in the abstract and text section.
Point 8: Line 144. The average age of pregnant women must be specified.
Response 8: Thank you for your advice. Following your suggestion, we have clarified this statement (Lines 146-147, page 4).
“The average age (mean ± SD) of participant women was 29.12 ± 3.14 years.”
Point 9: Lines 230 to 232: This expression must be explained in the discussion section.
Response 9: We sincerely appreciate your feedback. Following your suggestion, we have clarified the expression related to Table S3, which shows the results of the sensitivity analysis. This adjustment is now fully explained in the discussion section (Lines 367-369, page 11).
“Additionally, the findings from our sensitivity analysis reveal that our study is robust because our results mirror those of the total population, even after the exclusion of some variables.“
Point 10: Minor editing of English language required.
Response 10: Thanks for your suggestions, the writing skills have been improved. We have discussed the writing issues with a professional English supervisor and made the necessary adjustments in the manuscript. We feel sorry for any inconvenience caused during the review process and hope that the revised version may meet your expectations.

Reviewer 4 Report
Comments and Suggestions for Authors
Although the introduction is supported by relevant bibliographic sources, it should be integrated with general information on PFAS and related maternal-fetal diseases.
The collection of data in the tables presented is suitable for the study, it is recommended, to facilitate reading and the references indicated, to add the meaning of the acronyms present in the legend of the tables.
The methods used in the study are adequate, despite the presence of numerous data collected, the authors have summarized the presentation.
As indicated by the authors, the manuscript presents limitations resulting from the lack of data on the biological mechanisms attributable especially to PFAS mixtures and the type of contact.
The text needs to be rechecked for the presence of any typos
Comments on the Quality of English LanguageThe text needs to be rechecked for the presence of any typos
Author Response
Response to Reviewer 4 Comments
Point 1: Although the introduction is supported by relevant bibliographic sources, it should be integrated with general information on PFAS and related maternal-fetal diseases.
Response 1: We appreciate your suggestion. Following it, we have searched for 6 new articles about PFAS and maternal-fetal diseases and added them to the introduction section (Lines 48-51, page 2) . This revision has resulted in a more comprehensive and well-rounded understanding of PFASs and its associations with maternal-fetal diseases in our paper.
“During this critical period of growth, there is evidence of reduced birthweight [4,5], neurobehavioral development [6,7], weakened immune responses [8,9], liver function [10], and increased adiposity in neonates and children [11,12].”
- Fan, X.; Tang, S.; Wang, Y.; Fan, W.; Ben, Y.; Naidu, R.; Dong, Z. Global exposure to per-and polyfluoroalkyl substances and associated burden of low birthweight. Environmental Science & Technology 2022, 56, 4282-4294, doi:10.1021/acs.est.1c08669.
- Wikström, S.; Lin, P.-I.; Lindh, C.H.; Shu, H.; Bornehag, C.-G. Maternal serum levels of perfluoroalkyl substances in early pregnancy and offspring birth weight. Pediatric Research 2020, 87, 1093-1099, doi:10.1038/s41390-019-0720-1.
- Forns, J.; Verner, M.-A.; Iszatt, N.; Nowack, N.; Bach, C.C.; Vrijheid, M.; Costa, O.; Andiarena, A.; Sovcikova, E.; Høyer, B.B. Early life exposure to perfluoroalkyl substances (PFAS) and ADHD: a meta-analysis of nine European population-based studies. Environmental health perspectives 2020, 128, 057002, doi:10.1289/EHP5444.
- Carrizosa, C.; Murcia, M.; Ballesteros, V.; Costa, O.; Manzano-Salgado, C.B.; Ibarluzea, J.; Iñiguez, C.; Casas, M.; Andiarena, A.; Llop, S. Prenatal perfluoroalkyl substance exposure and neuropsychological development throughout childhood: The INMA Project. Journal of Hazardous Materials 2021, 416, 125185, doi:10.1016/j.jhazmat.2021.125185.
- Grandjean, P.; Andersen, E.W.; Budtz-Jørgensen, E.; Nielsen, F.; Mølbak, K.; Weihe, P.; Heilmann, C. Serum vaccine antibody concentrations in children exposed to perfluorinated compounds. Jama 2012, 307, 391-397, doi:10.1001/jama.2011.2034.
- Granum, B.; Haug, L.S.; Namork, E.; Stølevik, S.B.; Thomsen, C.; Aaberge, I.S.; van Loveren, H.; Løvik, M.; Nygaard, U.C. Pre-natal exposure to perfluoroalkyl substances may be associated with altered vaccine antibody levels and immune-related health outcomes in early childhood. Journal of immunotoxicology 2013, 10, 373-379, doi:10.3109/1547691X.2012.755580.
- Stratakis, N.; Conti, D.V.; Jin, R.; Margetaki, K.; Valvi, D.; Siskos, A.P.; Maitre, L.; Garcia, E.; Varo, N.; Zhao, Y. Prenatal exposure to perfluoroalkyl substances associated with increased susceptibility to liver injury in children. Hepatology 2020, 72, 1758-1770, doi:10.1002/hep.31483.
- Høyer, B.B.; Ramlau-Hansen, C.H.; Vrijheid, M.; Valvi, D.; Pedersen, H.S.; Zviezdai, V.; Jönsson, B.A.; Lindh, C.H.; Bonde, J.P.; Toft, G. Anthropometry in 5-to 9-year-old Greenlandic and Ukrainian children in relation to prenatal exposure to perfluorinated alkyl substances. Environmental health perspectives 2015, 123, 841-846, doi:10.1289/ehp.1408881.
- Mora, A.M.; Oken, E.; Rifas-Shiman, S.L.; Webster, T.F.; Gillman, M.W.; Calafat, A.M.; Ye, X.; Sagiv, S.K. Prenatal exposure to perfluoroalkyl substances and adiposity in early and mid-childhood. Environmental health perspectives 2017, 125, 467-473, doi:10.1289/EHP246.
Point 2: The collection of data in the tables presented is suitable for the study, it is recommended, to facilitate reading and the references indicated, to add the meaning of the acronyms present in the legend of the tables.
Response 2: Thank you for your suggestion. According to your suggestion, we have included the explanations for the acronyms used in the legend of the tables and figures.
Line 165, Page 4. “Abbreviations: LOQ, limit of quantification; DR, detection rate; GM, geometric mean.” was added.
Line 181, Page 6. “Abbreviation: OR, odds ratio; CI, confidence interval; Ref: Reference.” was added.
Line 189, Page 7. “Abbreviation: OR, odds ratio; CI, confidence interval; Ref: Reference.” was added.
Line 235, Page 9. “Abbreviation: OR, odds ratio; CI, confidence interval.” was added.
Point 3: The methods used in the study are adequate, despite the presence of numerous data collected, the authors have summarized the presentation.
Response 3: We appreciate your feedback. We have made an effort to ensure that the presentation of methods remains concise and appropriate, despite collecting extensive data. If you have any suggestions or concerns regarding our methods, we would like to address them.
Point 4: As indicated by the authors, the manuscript presents limitations resulting from the lack of data on the biological mechanisms attributable especially to PFAS mixtures and the type of contact.
Response 4: Thank you for your input. We appreciate your feedback and recognize the obstacles presented by limited data in current research on biological mechanisms. We have emphasized these limitations in the discussion section to make readers aware of the scope and applicability of our study (Lines 358-360, page 11).
“Second, the lack of data limits our full understanding of its impact and related biological mechanisms. Furthermore, further research is imperative for a comprehensive understanding.”
Point 5: The text needs to be rechecked for the presence of any typos.
Response 5: We have thoroughly reviewed the manuscript to identify and correct any typos or errors that may be present. And we have made the necessary revisions to improve the overall quality of the manuscript.

Round 2
Reviewer 3 Report
Comments and Suggestions for Authors
The work has been improved according to suggestions.